Optimizing a 3D convolutional neural network to detect Alzheimer’s disease based on MRI

http://orcid.org/0009-0007-1996-8088 Alarjani Maitha 1 metha2alarjani@gmail.com
Almuaibed Abdulmajeed 2
1 Computer Science, King Faisal University , Al-Ahsa , Saudi Arabia
2 Project Management, General Administration of Services and Facilities, King Faisal University , Al-Ahsa , Saudi Arabia
Coelho Paulo Jorge
Electronic publication date: 2025 Aug 29
Publication date: 2025
Volume: 11
Electronic Location ID: e3129
Received 2024 Nov 14; Accepted 2025 Jul 25
Copyright: © 2025 Alarjani and Almuaibed
Copyright year: 2025
Copyright holder: Alarjani and Almuaibed
License: This is an open access article distributed under the terms of the Creative Commons Attribution License, which permits unrestricted use, distribution, reproduction and adaptation in any medium and for any purpose provided that it is properly attributed. For attribution, the original author(s), title, publication source (PeerJ Computer Science) and either DOI or URL of the article must be cited.
License URL: https://creativecommons.org/licenses/by/4.0/

Keywords: Deep learning, 3D-convolutional neural networks, Data pre-processing, Medical image processing, Neuroscience, MRI volumetric data, Brain atrophy detection, Neurodegenerative disorders, Alzheimer’s disease classification

Funding: Deputyship for Research and Innovation Ministry of Education in Saudi Arabia KFU251759 This research was funded by the Deputyship for Research and Innovation, Ministry of Education in Saudi Arabia (KFU251759). The funders had no role in study design, data collection and analysis, decision to publish, or preparation of the manuscript.

==============================
Alzheimer’s disease (AD) is a progressive neurological disorder that affects millions worldwide, leading to cognitive decline and memory impairment. Structural changes in the brain gradually impair cognitive functions, and by the time symptoms become evident, significant and often irreversible neuronal damage has already occurred. This makes early diagnosis critical, as timely intervention can help slow disease progression and improve patients’ quality of life. Recent advancements in machine learning and neuroimaging have enabled early detection of AD using imaging data and computer-aided diagnostic systems. Deep learning, particularly with magnetic resonance imaging (MRI), has gained widespread recognition for its ability to extract high-level features by leveraging localized connections, weight sharing, and three-dimensional invariance. In this study, we present a 3d convolutional neural network (3D-CNN) designed to enhance classification accuracy using data from the latest version of the OASIS database (OASIS-3). Unlike traditional 2D approaches, our model processes full 3D MRI scans to preserve spatial information and prevent information loss during dimensionality reduction. Additionally, we applied advanced preprocessing techniques, including intensity normalization and noise reduction, to enhance image quality and improve classification performance. Our proposed 3D-CNN achieved an impressive classification accuracy of 91%, outperforming several existing models. These results highlight the potential of deep learning in developing more reliable and efficient diagnostic tools for early Alzheimer’s detection, paving the way for improved clinical decision-making and patient outcomes.

Introduction

The brain is a dynamic, complicated organ that controls many things, like emotions, memory, creativity, and intelligence (Miah et al., 2021). The brain is made of the cerebrum, cerebellum, and stem of the brain and shielded by the skull (Goedert & Spillantini, 2006). Many diseases cause brain damage, such as epilepsy, Tumours, sleep disorders, and Alzheimer’s disease (AD). AD is a well-known neurological problem that consistently deteriorates an individual’s memory and cognitive function. It is persistent and cannot be reversed. Cerebral cortex atrophy progresses in AD, resulting in irreversible brain damage (Scheltens et al., 2016).

In 2019, around six million people were impacted by AD, and it is expected to reach 14 million by 2050 (Alzheimer’s Association, 2018). Because the prodromal phase of AD is long, it primarily affects the elderly (Bhagwat et al., 2018). AD is not curable but can be managed by taking drugs and doing physical activity. The early identification of disease is important so that the patient life’s quality is not affected. It is projected that early detection and intervention might save around $7.9 trillion because earlier initiation of medicine can keep patients functional for a longer period and reduce costs associated with disability (Bhagwat et al., 2018).

With the development of medical imaging technology, image analysis now offers adequate detail about the subject under observation. Learning patterns can detect cerebral neurodegeneration by utilizing the information included in the images. As AD advances, due to different modalities such as positron emission tomography (PET), magnetic resonance imaging (MRI), and computed tomography (CT) may quantify cerebral and hippocampus atrophies as well as ventricular widening (Kasban, El-Bendary & Salama, 2015).

There is a significant gap between health care specialists’ and researchers collaboration on the path of investigation and discovery from clinical trials. Neurologists utilize magnetic resonance imaging (MRI) to visualize any neuronal abnormalities in the brain linked with Alzheimer’s disease. It is a non-invasive, extensively used, and time-tested method (Young, 2007). A T1-weighted magnetic resonance image (T1-MRI) is chosen for an accurate evaluation of structural alterations since it offers structural changes with high resolution. MCI and CN stand for mild cognitive impairment and cognitive normalcy, respectively. MCI is the pre-clinical stage of AD. MCI patients are more likely to acquire AD.

Technological gap

There is a need for advanced tools and techniques that could be utilized to identify AD and to overcome the subsequent hurdles in providing improved treatment procedures. We need to find a good architecture for this problem to achieve outstanding performance. The deep learning technology in the Squeezenet model might give reasonable performance on 3D MRI images. Furthermore, we need to find the best hyperparameter for this model.

The proposed study’s outcomes can be summarized as follows: Collect MRI images from reputable medical sources and databases.

Use image augmentation for the data to increase its size and add robustness.

Deep learning on-the-fly was used to extract key features from MRI scans.

Propose a deep 3D convolutional neural network architecture, SESENet18.

This article begins with an overview of Alzheimer’s disease (AD) and examines key hypotheses, specifically the cholinergic and amyloid hypotheses. A comprehensive literature review highlights recent research on AD and identifies existing gaps in the field. The materials and methods section outlines the structure and approach of the proposed system. Next, the results and discussion analyze its performance, followed by a comparison with recent techniques to highlight its strengths. The article then discusses the advantages of the proposed method in detail. Finally, it concludes with key findings and suggests potential directions for future research.

Hypotheses for AD

Cholinergic hypothesis

In the 1970s, the enzyme choline acetyltransferase (ChAT) key to acetylcholine (ACh) production was found to be involved in cholinergic abnormalities affecting both neocortical and pre-synaptic areas. Given ACh’s essential role in brain function, a cholinergic theory of AD was proposed. ChAT generates ACh in the cytoplasm of cholinergic neurons by combining choline with acetyl-coenzyme A, after which the vesicular acetylcholine transporter (VAChT) directs it to nerve cells. ACh is fundamental to many brain processes, including learning, concentration, sensory processing, and memory. The deterioration of cholinergic neurons, which leads to memory loss and cognitive decline, has been linked to the progression of AD.

Studies suggest that amyloid- β oligomers and interactions between acetylcholinesterase (AChE) and amyloid peptides are associated with cholinergic synapse loss and amyloid fibril formation. Additionally, deficits in nicotinic and muscarinic (M2) ACh receptors at pre-synaptic cholinergic junctions, along with reduced excitatory amino acid (EAA) activity—marked by decreased glutamate and D-aspartate levels in various cortical areas—are also linked to AD development. Furthermore, cholinergic receptor antagonists such as scopolamine, known to induce memory loss, are sometimes utilized in research. To counteract these effects, substances that enhance acetylcholine synthesis are employed (Norfray & Provenzale, 2004).

Amyloid hypothesis

The amyloid hypothesis was proposed after the discovery that the accumulation of aberrant amyloid plaques, particularly those formed by amyloid-beta peptides, was strongly correlated with neurodegenerative changes in the brain, particularly in the context of mild cognitive impairment (MCI). It was observed that, with aging, even healthy individuals experience some degree of amyloid plaque deposition. This led to the hypothesis that the accumulation of these plaques might play a crucial role in the development of AD. While alternative theories have been put forward to explain non-inherited forms of AD, such as the non-inherited Alzheimer’s disease (NIAID), the amyloid hypothesis remains the most widely accepted explanation for the pathogenic mechanism underlying inherited AD (IAD). According to this theory, aging or other pathological conditions impair the function of alpha-secretase, an enzyme responsible for degrading amyloid precursor proteins, which in turn leads to an increase in the concentration of amyloid-beta peptides (A40 and A42), thus promoting the development of AD (Alzheimer’s Association, 2019).

Both the cholinergic and amyloid hypotheses highlight the multifactorial nature of AD, with amyloid plaque accumulation and cholinergic dysfunction being key drivers of cognitive decline. These mechanisms emphasize the need for more accurate and efficient classification models. By incorporating these factors into diagnostic models using advanced machine learning (ML) and deep learning (DL) techniques, we can improve the sensitivity and specificity of AD detection. Enhanced models could better capture the complex interactions between amyloid plaques, cholinergic dysfunction, and other factors, leading to earlier detection and more personalized treatments.

Literature review

Many methods have been proposed to recognize AD. This section reviews studies that used traditional ML and DL techniques for AD diagnosis. Early approaches focused on analyzing brain anatomical and structural images, like MRIs, treating the task as a classification problem. These methods relied heavily on manually designed features, such as voxel, region, and patch-based representations. However, training classification models required time-consuming manual image segmentation (Cuingnet et al., 2011).

Liu et al. (2015) came up with a method for classifying AD/MCI called inherent structure-based multiview learning (ISML). The proposed methodology has three steps: Grey matter (GM) tissues are used as tissue-segmented brain images for (1) multiview extracting features utilizing multiple templates, (2) feature selection employing voxel selection-based subclass clustering to enhance the power of characteristics, and (3) ensemble classification using support vector machine (SVM). On the MRI initial dataset of 459 participants from the ADNI database repository, they evaluated the effectiveness of the suggested technique. The experiment’s findings indicate that for AD vs. normal control (NC), the suggested ISML approach achieves an accuracy of 93.83%, precision of 95.69%, and sensitivities of 92.78%. Consequently, Krashenyi et al. (2016) used fuzzy logic to classify AD based on PET and MRI datasets from the ADNI database, which included 818 subjects (229 healthy controls, 401 mild cognitive impairment, and 188 AD subjects). Their method consists of three steps: image pre-processing, feature selection based on region of interest (ROI) using t-tests, and fuzzy categorization using the c-means method. The classification performance was measured using the area under the curve (AUC), with the best AUC of 0.99 achieved for the training set. The method achieved an AUC of 0.8622 for AD vs. healthy controls. Lazli, Boukadoum & Mohamed (2018) presented a computer-aided analysis (CAD) scheme for distinguishing between AD cases and normal controls using MRI and PET images. The method consists of two steps: segmentation and classification. They employ fuzzy possible tissue segmentation, combining fuzzy c-means (FCM) and possibilistic c-means (PCM), followed by classification using SVM models with various kernels (linear, polynomial, and radial basis function (RBF)). The proposed method was tested on MRI and PET scans of 45 individuals with Alzheimer’s and 50 healthy individuals from the ADNI dataset using leave-one-out cross-validation. Results showed better performance as compared to recent methods, with 75% of accuracy for MRI images and 73% for PET images.

Krishna Thulasi & Varghese (2018) did similar work and proposed an Alzheimer’s diagnosis system founded on image processing methods and support vector machine (SVM) models. The projected method was used for training and testing on a minor MRI scan from the ADNI database, which had 70 AD and 30 NC subjects. The projected solution had two parts: extracting/selecting features and grouping them. In the primary step, the writers used speeded-up animated features (SURF) to pull out the most important parts of the matching MRI pictures. After that, they used the grey-level co-occurrence matrix (GLCM) to pull out the attributes. During the classification stage, they used SVM to put MRI scans into two groups: those with AD and those with normal controls. Mehmood et al. (2020) developed a 2D Siamese convolutional neural network (CNN) based on the visual geometry group (VGG) structure. They presented convincing results after evaluating their model using the OASIS dataset’s training and testing split. However, based on their proposed data flow diagram, it is likely that they inadvertently introduced data leakage in their preprocessing method through the use of augmentation techniques.

In a recent study (Korolev et al., 2017), deep learning was applied to AD using 3D convolutional neural networks (CNNs) based on scaled-down versions of VGG and residual network (ResNet) architectures. However, only binary classification was performed. The study successfully extracted cortical density, surface energy, volumetric measures, and texture features from the CAD Dementia competition dataset. A stacked autoencoder with three hidden layers was trained using these features, achieving successful performance. In another study (Abrol, 2020), authors used 3D CNNs based on deep networks for multiclass and binary tasks in AD. They created a training dataset for cross-validation and a smaller test set using ADNI data. While their results were encouraging, they did not compare with other databases or standardized rating frameworks. Additionally, they encountered fitting problems in training examples.

Research by Rallabandi et al. (2020) proposed a design for early detection and classification of AD and MCI from normal aging, as well as predicting and diagnosing early and late MCI. Using the ADNI database, they analyzed 1,167 whole-brain MRI subjects, including 371 normal controls (NC), 328 early MCI, 169 late MCI, and 284 AD. FreeSurfer analysis was used to extract 68 attributes related to the cortical layer width from each MRI image. These features were used to build a model, which was evaluated using various ML methods such as non-linear SVM (RBF kernel), naive Bayes, k-nearest neighbors, random forest, decision tree, and linear SVM. The model achieved 75% accuracy in distinguishing all four groups with the highest receiver operating characteristic area under the curve (ROC AUC).

Ji et al. (2019) proposed a technique for the initial diagnosis of AD using MRI scans and convolutional neural networks (ConvNets) was developed. They used 615 MRI images from ADNI, categorized into (179 AD, 254 MCI, and 182 NC). The MRI scans were resized to 224 × 224 and divided into white matter (WM) and gray matter (GM) groups. Twenty portions of GM and WM datasets were selected and used to train the ConvNet model. Group learning methods were employed after convolutional operations to enhance classification. ResNet50, NASNet, and MobileNet were chosen as the best classifiers. The method achieved a 98.59% accuracy for AD vs. NC, 97.65% of accuracy for AD vs. MCI, and 88.37% of accuracy for MCI vs. NC. Castellazzi et al. (2020) performed MRI data analysis using the ML methods. AD and vascular dementia (VD) are two of the most common and may share multiple symptoms that can mislead the diagnosis. This problem is addressed by implementing a method for the classification of AD and VD. The ANN, neuro-fuzzy inference, and SVM are utilized to assess the method’s performance.

Zaabi et al. (2020) proposed a DL method for brain image classification into normal images or AD images using the Oasis dataset, the proposed method includes two primary steps: region of interest extraction and image classification utilizing two DL techniques: CNN and transfer learning. The transfer learning resulted in a higher accuracy of 92.86%. Basheer, Bhatia & Sakri (2021) Proposed novel capsule network for AD detection based on MRI and PET images and determine the significance of traits through correlation analysis between factors, and density of data using exploratory data analysis that was done using the (373 X 15) OASIS dataset to classify the labels into a person with dementia and person without dementia groups. They used many DL classifiers with various performance metrics. 92.39% accuracy was obtained. Castellano et al. (2024) evaluated both unimodal and multimodal imaging, including PET scans, for classification to leverage multimodal data from the same patient. Additionally, they incorporate 2D and 3D MRI scans for assessment. Integrating multiple modalities enhances overall model performance compared to using unimodal data. The proposed approach is validated on the OASIS-3 cohort, with explainability analysis performed using Grad-CAM.

Momeni et al. (2025) utilized sMRI and AI to classify AD, early MCI (EMCI), late MCI (LMCI), and NC. A total of 398 participants from the ADNI and OASIS databases were analyzed, comprising 98 AD, 102 EMCI, 98 LMCI, and 100 NC cases. The proposed model achieved outstanding performance with 99.7% accuracy and high AUC values: NC vs. AD (0.985), EMCI vs. NC (0.961), LMCI vs. NC (0.951), LMCI vs. AD (0.989), and EMCI vs. LMCI (1.000). By leveraging DenseNet169, transfer learning, and class decomposition, the model effectively classified AD stages, with a notable ability to differentiate EMCI from LMCI, aiding in early diagnosis and intervention. Tuncer, Dogan & Subasi (2025) introduced FiboNeXt, a lightweight CNN inspired by the Fibonacci sequence and ConvNeXt architecture for AD detection. Using a public MRI dataset with augmented and original versions, the model achieved high accuracy: 95.40% and 95.93% (validation) and 99.66% and 99.63% (testing) across two datasets. The results demonstrated FiboNeXt’s effectiveness and potential for broader computer vision applications.

Despite significant advancements, challenges persist in brain MRI analysis, particularly due to the complexity of brain anatomy and the diverse manifestations of AD. Addressing these challenges requires AI models tailored for neurodegenerative disorders. This research presents a tailored 3D convolutional neural network (3D-CNN) model developed for the detection of Alzheimer’s disease using MRI scans. The proposed architecture enhances conventional deep learning approaches by incorporating optimized feature extraction and learning strategies, specifically adapted to the complexities of brain MRI data. By addressing these domain-specific challenges, our method advances the application of AI in neuroimaging and offers a promising direction for improving the accuracy and reliability of Alzheimer’s disease diagnosis.

Materials and Methods

The proposed approach marks a notable step forward in applying 3D-CNNs to the analysis of MRI scans for Alzheimer’s disease. This enhanced architecture is purpose-built to tackle the complex nature of AD classification, harnessing the power of deep learning to improve both the accuracy and efficiency of the diagnostic process. Figure 1 presents the model’s workflow, offering a clearer depiction of its operational mechanism.

Figure 1 The proposed model workflow involves collecting MRI data, performing image preprocessing, applying 3D augmentations, extracting meaningful features, and classifying the data using a 3D-CNN based on the SE-ResNet-18 architecture.

Image pre-processing

Feature extraction is a challenging task due to images complex nature. All pre-processing techniques are essential for the model during training, to make the images clearer and more useful then reach a good performance. Also, it’s necessary during deployment as a part of the model to make the input shape of the images match the input shape of the model and also to reach a good testing performance. The size of the images in the dataset differs. The first step is to change their shape, so they are all the same size. All the parts of our design model must be the same size, and if the 3D images are too big, the design might not work (Vani Kumari & Usha Rani, 2020). Images are resized to 128 × 128 × 32 to balance computational efficiency and retain critical features, ensuring enough detail for accurate analysis while reducing processing load.

3D data augmentation (Aug)

In data augmentation, the data is enhanced to obtain more information. Enhancing the dataset may involve making small adjustments to the data or utilizing machine learning algorithms to produce new points of information in the subspace of the original information. Most of the time, the more data there is, the better neural networks work. One of the biggest problems with training good models is that the training data are not always available, are poor quality, or are not labeled. Without any need to gather more data, computer vision data augmentation is a powerful method for enhancing the efficiency of our machine vision algorithms. We build new versions of our photographs from the originals while purposefully adding flaws. Instead of forcing our model to memorize the exact ways things appear in our training examples, this allows it to understand what an entity typically looks like. One common way to get more data is to make small changes to our data. CNN work the same no matter how they are moved, viewed, sized, or lit. Image data augmentation changes images in the training data to fit into the same class as the original picture. In this method, common bases of difference are added to the training examples, which can help to keep neural network models from becoming too accurate. Over-fitting occurs when a model can fit training data flawlessly but does not do well with new data because it just remembers the training. Transformations need to be carefully chosen based on the training dataset and what is known about the problem domain (Chlap et al., 2021). Random rotation: In this method, the images are rotated or the position is altered with respect to the original image. This includes random rotation with clockwise or anti-clockwise. If there is a fail in the object detection, the bounding box should also be altered according.

Random crop: It produces a random portion of the original picture. This helps with generalizing the systems and helps the model to learn if the region of interest is invisible.

Flip: Images can be turned sideways or up and down. A vertical flip is the same as turning an image 180 degrees and then doing a horizontal flip by switching the pixels in the rows and columns.

Crop and change size: A random part of the original image is taken, and then that part is resized to match the size of the original image. Since CNNs do not change when translated or shrunk, this can help make the model more stable.

Rotation: Adding a random rotation to the image is another way to change it. Depending on the image, a rotation can cause artifacts in areas where new information needs to be added after the rotation. There are many ways to fill, like adding zeros, reflecting, wrapping, etc. Since the background of an MRI is always black (zero), this problem does not happen.

Feature extraction

Deep learning, a rapidly emerging area within machine learning, is gaining significant attention in the analysis of large-scale, high-dimensional medical imaging datasets. In our approach, we utilized deep learning to analyze large-scale, high-dimensional medical imaging datasets. By leveraging raw neural imaging data, our model automatically generates features through ‘on-the-fly’ learning, which allows for more accurate and efficient analysis. We focused on ensuring that our model could effectively work with real-world images, utilizing extensive, balanced training data to capture the complexity of the data. This approach enables the model to deliver high performance in tasks such as disease detection and classification, benefiting from the strengths of deep learning in handling intricate medical imaging challenges (Wang et al., 2020).

3D convolutional networks

In order to encode the quality and reduce the number of hyper-parameters, 3D-CNNs are specifically constructed based on the process of ensuring that the raw data are two-dimensional (pictures). The 3D-CNN topology builds on conventional feed-forward back propagation learning by utilizing spatial structures to reduce the number of variables that must be acquired. The 3D-CNNs transform the input taken from the input layer into a class scores collection provided by the output layer by using all fully connected. The 3D-CNN structure has various versions (Satya, 2020).

The input layer socially accepts three-dimensional information in the shape of the image’s size (width × height) and has color streams represented by depth. A common 3D-CNN model typically consists of convolutional and pooling layers that are alternated and succeeded by one or more completely connected layers. The layer occasionally takes the role of the fully linked layer is referred to as global average pooling. To improve 3D-CNN performance, several regulatory units, including regularization and washout, are also implemented along with various adaptive thresholding. Developing novel architectures and obtaining enhanced results depend critically on the configuration of 3D-CNN elements. The function of these elements in a 3D-CNN structure is covered in this subsection. Convolution layer: The fundamental layer of a convolutional algorithm CNN architectural elements. Convolutional layers use a region of directly connected neurons from the preceding layer to alter the input data layer. The layers will calculate a dot product here among the input layer’s input neurons’ local scores, and their region in the output layer is linked. Pooling layer: Layers that offer a method for this purpose. By briefing the existence of attributes on coverings of the feature map, down-sampling feature maps are accomplished. By combining the outputs of neuron groups at a single layer into a single neuron, pooling layers reduce the data dimensionality (Shin et al., 2016). The architecture of the model is shown in Fig. 2.

Figure 2 Squeeze-and-excitation residual network-18 architecture of the proposed model: The proposed model integrates squeeze-and-excitation (SE) blocks into a ResNet-18 backbone to enhance feature recalibration.

This architecture effectively captures channel-wise dependencies and boosts the network’s representational capacity, improving the classification of Alzheimer’s disease using 3D MRI data.

Squeeze and excitation net

Squeeze-and-excitation residual network (SqueezeNet) is a DNN framework that can obtain accuracy similar to AlexNet on ImageNet with 50 times fewer parameters. Furthermore, the compression techniques can reduce the size to less than 0.5 MB which is 510 times smaller than (AlexNet) (Iandola et al., 2016). The ResNeXt and squeeze-and-excitation network (SENet) are combined to form the SE-ResNeXt, which this research uses for nodule categorization. A highly modular design network for classification tasks is called ResNeXt. It differs from ResNet in that it includes the idea of “attribute values,” which is crucial to the classification process. It is common knowledge that a deep neural network’s depth and width significantly affect its ability to represent information. The deterioration issue will, however, become apparent as the network grows deeper or wider and the disappearing completely gradients problem is shown. The depth and width may begin to provide declining returns in this situation. Luckily, the ResNeXt successfully avoids this issue and investigates an effective strategy for raising classification accuracy, namely, raising cardinality (see Figs. 3, 4).

Figure 3 The ResNeXt model enhances feature learning by increasing cardinality (independent paths within blocks), as inspired by Iandola et al. (2016), improving the model’s ability to capture diverse features efficiently.

Figure 4 SENet module inspired by Gu et al. (2019).

The excitation procedure is used to effectively capture channel-wise connections and utilize the aggregated data from the squeeze process. This is achieved through a gating system, as explained in Eq. (1).

(1) S=FEX(Z,W)=α(W2δ(W1Z)).

It is more efficient to design the deep network to increase the feature discrimination ability of lung nodules thanks to the 3D SE-ResNeXt component. Additionally, it incorporates the benefits of feature recycling by choosing to boost informative nodule traits and repressing fewer effective ones. As a result, the 3D SE-ResNeXt employed is trustworthy and efficient at automatically acquiring high discriminant information to classify AD and NC from MRI images (Gu et al., 2019).

Layer of convolutions

The convolutional layer is the most important part of the DL CNN and the main building block. It is in charge of the process of extracting features and also makes feature maps. The convolutional is based on a static quantity of filters that detects the s attribute and pulls out the features by combining the filters and input image. During training, each filter finds low-level attributes like colors, boundaries, blobs, and angles in the analyzed images.

Pooling layer

After the convolutional layers, there are the pooling layers (Conv). The sub-samples in a layer are in charge of making the feature maps used to make the convolutional layers smaller. Max pooling decreases the attribute maps by removing the small part of the pictures with the highest value. This prevents overfitting by summarizing the parts of the image that represent it. It also minimizes the parameters, which lowers the amount of work that needs to be done on the computer. Pooling is also another name for the average pooling layer. This layer works like max pooling, but instead of taking the maximum value, it calculates the averages of two-by-two rectangles to make a subsampled image.

Layer for batch normalization

The batch normalization layer normalizes the output of the convolution layer by setting the batch’s value of the mean to 0 and the variance value to 1. Using higher learning rates, this method speeds up the learning process. Also, it keeps the model’s gradients from going away during backpropagation. The batch normalization layers are also more resistant to the wrong initialization of weights.

Dropout layer

To overcome the overfitting the dropout layer is utilized. This method is based on how neurons are taken out randomly during training. The dropout rate parameter controls how many neurons are lost, determining how likely a neuron will be lost. Only during the training process are the neurons taken away.

Fully connected layer

The fully connected layer serves as the final stage of the network, playing a crucial role in connecting the preceding layers and producing the ultimate classification output. This layer acts as a classifier, synthesizing the learned features from the network to make predictions. Typically, a softmax activation function is employed in the final layer, which transforms the output into a probability distribution, ensuring that the predicted class is based on the most probable outcome (Karasawa, Liu & Ohwada, 2018).

Results and discussion

Dataset

The OASIS (Open Access Series of Imaging Studies) dataset is a publicly available collection of neuroimaging and clinical data aimed at supporting AD research and the study of normal aging. The dataset includes neuroimaging data obtained from 1,098 individuals, ranging from 42 to 95 years of age. Of these, 605 participants are classified as NC, and 493 participants have varying levels of cognitive decline associated with AD (Marcus et al., 2007). The data was collected over 15 years by the Washington University Knight AD research center. It comprises a variety of imaging modalities, including approximately 2,000 MR sessions, which span both structural MRI and functional fMRI techniques. The dataset is longitudinal, incorporating not only neuroimaging data but also cognitive, clinical, and biomarker information, making it a rich resource for AD research.

The OASIS-3 dataset is divided into two main classes: NC: Consists of 930 3D MRI images from healthy adults.

AD: Includes 945 3D MRI images from participants with various stages of cognitive decline, ranging from very mild cognitive impairment (MCI) to moderate and severe dementia.

All data in the OASIS dataset is T2-weighted MRI, a specific imaging type used for brain structure visualization. This comprehensive dataset is widely used for the development and evaluation of algorithms and tools aimed at improving early diagnosis and understanding of AD and normal aging processes, as shown in Table 1.

Table 1 OASIS MRI dataset description.

Class	# of subjects	# of scan after Aug	Age range	Data type	
NC	605	930 3D images	50–80 years	T2 MRI	
AD	493	945 3D images	42–95 years	T2 MRI	
Total	1,875 3D images	

Evaluation metrics

In this work, we use accuracy, recall, precision, and F1-scores to measure the performance of a model. Accuracy and the F1-score are calculated based on true negatives (TN), true positives (TP), and false positives (FP). Precision measures the quality of the positive predictions computed by the model and is calculated as the number of true positives over the number of positive predictions, which includes both true and false positives.

(2) Precision=TPTP+FP

Recall, also referred to as sensitivity (SN) is the model’s capability to identify AD patients and is given as follows:

(3) Recall=TPTP+FN

F1-score offers both the precision and recall of the test to calculate the score and can be computed as follows:

(4) F1-score=2⋅TP2⋅TP+FP+FN

Accuracy (ACC) is the likelihood of correct positive and negative forecasts.

(5) Accuracy=TP+TNTP+TN+FP+FN

where the parameters FN, FP, TP, and TN are as follows: TP refers to a subject who has AD and is also classified as AD.

In FP, the person is normal but classified as positive.

TN refers to the person who is normal and also classified as Normal.

In FN, the person has AD but is classified as normal.

Parameters play a critical role in defining a model’s behavior during training. These include choices such as the loss function, optimizer, learning rate, and evaluation metrics—all of which significantly impact how effectively the model learns from the data and generalizes to unseen examples. Selecting optimal parameters is essential for maximizing model performance, ensuring convergence, and avoiding overfitting, as shown in Table 2.

Table 2 Explanation of model compilation and training parameters.

Parameter	Description	
Loss function	Binary crossentropy, used for binary classification tasks.	
Optimizer	Learning rate = 1.0, optimization algorithm with adaptive learning rate.	
Validation data	Validation_data = validation_dataset—Used to validate the model after each epoch.	
Epochs	50, number of times the model will go through the entire training dataset.	
Shuffle	True-Shuffles the data before each epoch to avoid order bias.	
Verbose	2-Displays one line of progress per epoch.	
Callbacks	Checkpoint cb—A callback to save the model at specific intervals.	

In our experiment, we used augmented 3D MRI images to train and evaluate the model. The dataset was randomly split into 85% for training and 15% for testing to ensure a balanced and unbiased evaluation. This corresponded to approximately 1,589 images used for training and 281 images for testing. Data augmentation techniques were employed to increase dataset diversity and improve the model’s robustness. The model was trained and evaluated over 50 epochs. This number was chosen to allow for thorough learning across all training instances and to observe the model’s performance trend over time. Training with 50 epochs resulted in consistently improved accuracy and reduced loss, indicating that the model was learning effectively without signs of overfitting.

The final results demonstrated a test accuracy of 91% and a low test loss of 0.503, suggesting that the model performed well on previously unseen data. This high accuracy, combined with the low loss value, highlights the effectiveness of both the model architecture and the preprocessing strategy. Performance trends during training and validation are illustrated in Figs. 5, 6, and 7, which present the model’s loss curve, accuracy progression, and the receiver operating characteristic (ROC) curve, respectively. These visualizations further confirm the stability and effectiveness of the proposed approach.

Figure 5 Loss of training and validation of the model.

Figure 6 Accuracy of training and validation of the model.

Figure 7 ROC curve.

In Fig. 7 The visual representation of the outcomes of the model to correctly classify data as TP against the FPR at various thresholds is given with the help of the ROC curve. The AUC quantifies the overall classification performance into a single metric. A model with a high AUC value (close to 1) demonstrates strong discriminative power, while the approximate value of 0.5 suggests random guessing performance.

Table 3 shows The classification report that our model performs well based on all measurements. Also, there is no significant overfitting as both training and validation data results are near each other. Also, there is no bias toward specific classes; the reason can be that both classes have almost the same number of images, making them balanced.

Table 3 Classification report.

	Precision	Recall	Sensitivity	F1-score	Specificity	
NC	98%	100%	95%	91%	82.1%	
AD	100%	97%	87%	90%	87%	
Accuracy	91%	

The Wilson Score Interval calculates more accurate confidence intervals for proportions, especially with small samples or extreme values. Confidence intervals represent the range in which a population parameter is likely to fall, with a specified confidence level (Balayla, 2024).

Comparison with state-of-the-art methods

To evaluate the effectiveness of our model, we compared its performance with the results reported in Khagi et al. (2019), where the authors utilized pre-trained models to classify NC and AD using MRI images. Specifically, AlexNet consists of 25 layers, GoogLeNet comprises 144 layers, and ResNet50 is built with 177 layers. The study explored both shallow and fine-tuning approaches, analyzing the impact of different layer sections on classification performance. Their dataset included 28 NC and 28 AD patients, with 30 representative slices selected per subject, and was randomly split into training, validation, and testing sets using a 6:2:2 ratio. These models were trained on the OASIS dataset, the same dataset used in our study. The results of their experiments are summarized in Table 3. The Fig. 8 and Table 4 presents a plot of 95% confidence intervals (CIs) for classification metrics in AD detection. It compares two classes: NC in blue and AD in red across five performance metrics—precision, recall, sensitivity, F1-score, and specificity. Each point represents the mean value of the corresponding metric, with error bars indicating the confidence range. The NC class generally shows higher recall, while the AD class maintains strong precision. The plot provides a clear visualization of performance variability, ensuring a statistically robust assessment of the model’s accuracy. Table 5 illustrates that their highest-performing model achieved an accuracy of 94.6%. While this slightly surpasses our model, it comes at the cost of significantly higher computational time, requiring approximately 1 min and 25 s per 5 epochs—equating to a total training time of around 14.2 min. In contrast, our model achieves competitive accuracy while demonstrating superior computational efficiency. Additionally, our model outperforms other benchmarked architectures in both accuracy and training speed, highlighting its effectiveness for AD classification.

Figure 8 Plotting of 95% confidence intervals (CIs) for classification metrics in AD detection.

Table 4 95% confidence intervals for AD classification metrics.

Metric	NC class (n = 930)	AD class (n = 945)	
Precision	96.88–98.72%	99.60–100.00%	
Recall	99.59–100.00%	95.71–97.91%	
Sensitivity	93.40–96.22%	84.71–89.00%	
F1-score	88.99–92.67%	87.92–91.75%	
Specificity	79.51–84.43%	84.71–89.00%	

Table 5 Comparison of literature results with our approach.

Models	Total units	Layer weight transferred	Accuracy (%)	Cohen-Kappa (%)	Training time (min: s) for 5 epochs	
AlexNet (Lombardi et al., 2020)	25 layers	1:5	94.64	89.29	1:25	
1:11	93.15	86.31	1:17	
1:16	91.96	83.93	1:11	
1:22	84.82	69.64	0:51	
GoogLeNet (Marcus et al., 2010)	144 layers	1:33	88.99	77.98	3:21	
1:62	88.39	76.79	3:13	
1:104	82.74	65.48	2:06	
1:141	79.17	58.33	1:40	
ResNet50 (Marcus et al., 2010)	177 layers	1:43	94.35	88.69	6:59	
1:91	93.75	87.50	6:35	
1:134	92.56	85.12	4:38	
1:174	91.67	83.33	2:54	
Our approach	25 layers	1:5	91.00	85.00	0:45	
Note:

Bold represent the key results in our study that make it different from other studies.

Strengths of the proposed method

In this study, we used 3D SqueezeNet, a CNN deep learning model, to diagnose AD from MRI images. SqueezeNet is a small DNN architecture that achieves AlexNet-level accuracy on ImageNet with 50× fewer parameters. Additionally, with model compression techniques, we were able to compress SqueezeNet to less than 0.5MB (510× smaller than AlexNet). For nodule categorization, this research integrates ResNeXt and SENet, forming the SE-ResNeXt model. ResNeXt is a highly modular design network for classification tasks, differing from ResNet by incorporating the concept of ‘attribute values,’ which is crucial for the classification process. It is well-established that the depth and width of a deep neural network significantly influence its ability to represent information. However, as the network grows deeper or wider, issues such as deterioration and vanishing gradients become apparent. In such cases, depth and width may provide diminishing returns. Fortunately, ResNeXt effectively avoids these issues by exploring a strategy known as cardinality, which enhances classification accuracy.

To address the challenges posed by the large size of 3D MRI scans and the need for increased computational resources, we employ image augmentation techniques. By applying various transformations to the selected 2D slices, such as rotations, translations, scaling, and flipping, we artificially expand the dataset. This not only increases the size of the training data but also improves the model’s robustness by exposing it to a greater variety of input patterns. Image augmentation helps mitigate the risk of overfitting, enhances the generalization ability of the model, and allows for more effective training without requiring additional computational resources or storage space.

Discussion

Ethical and clinical considerations

While the proposed 3D CNN-based model shows promise in aiding the early diagnosis of AD, its deployment in clinical settings requires careful consideration. Transparency and explainability must be prioritized to ensure that healthcare professionals understand the rationale behind predictions, thus reducing the risk of bias or misinterpretation. AI systems should act as decision-support tools rather than replacements for clinical expertise. Ethical implementation also involves securing informed consent for the use of medical data, ensuring strict patient privacy protection, and adhering to data governance and regulatory standards. Before real-world adoption, such models must undergo extensive clinical validation, independent peer review, and continuous monitoring to ensure safety, reliability, and trustworthiness in medical practice.

Conclusions and future work

In this study, 3D-CNN approach was proposed to classify the normal MRI brain images of Alzheimer’s patients from abnormal images. Our study distinguishes among several deep neural network research on Alzheimer’s illness. In addition, most of the research focused on conventional pre-processing, feature extraction, and neural network methods, all of which generate high results compared to the typical, predictable procedures conducted manually by a regular medical expert. In addition, several researchers have exerted significantly more effort to diagnose AD utilizing the computer-assisted diagnosis method, which can diagnose the condition. In recent years, the study and interpretation of medical images have become one of the most important biomedical and bio-informatics engineering research fields.

Using AI-based tools, clinicians have achieved their primary aims for the automated early-stage diagnosis of AD. For the early diagnosis of AD patients, an automated framework and classification for AD based on MRI images is essential. This project developed a 3D-convolutional neural network classification approach for AD utilizing MRI data. Out of all the results with different epochs, only the 50-epoch outcome was statistically significant, with a 91% accuracy rate. Although the recommended technique has only been examined on a dataset about AD, we believe it may be successfully extended to other medical classification issues. The suggested model may be evaluated using other AD datasets and data from local hospitals and other neurological disorder diagnoses.

To improve the model’s generalization capabilities in future work, we can integrate additional MRI data and explore various image augmentation methods to generate diverse training samples, thereby enhancing the model’s robustness. Additionally, experimenting with different feature extraction techniques and model architectures will allow us to assess their impact on performance and identify the most effective configuration. Moreover, we aim to integrate the AD framework with ChatGPT, enhancing its diagnostic capabilities by incorporating advanced machine learning models and clinical guidelines, which could improve its accuracy and adaptability for personalized AD detection and early intervention strategies.

Supplemental Information

Supplemental Information 1 Model code.

Supplemental Information 2 Author Contribution Statement.

Additional Information and Declarations

Competing Interests

The authors declare that they have no competing interests.

Author Contributions

Maitha Alarjani conceived and designed the experiments, performed the experiments, analyzed the data, performed the computation work, prepared figures and/or tables, authored or reviewed drafts of the article, data Curation, File Organization, Resource Identification, and approved the final draft.

Abdulmajeed Almuaibed conceived and designed the experiments, performed the experiments, prepared figures and/or tables, authored or reviewed drafts of the article, project Supervision, Quality Control, performed computation, and approved the final draft.

Data Availability

The following information was supplied regarding data availability:

Code is available in the Supplemental Files, GitHub, and Zenodo: https://github.com/metho12/Optimizing-a-3DCNN.git.

Alarjani, M. (2025). 3D-CNN for Alzheimer’s Detection via MRI. Zenodo. https://doi.org/10.5281/zenodo.15448526.

Open Access Series of Imaging Studies (OASIS) is available at WashU Medicine: https://www.oasis-brains.org/. Request for access can be obtained at: https://sites.wustl.edu/oasisbrains/home/access/.

OASIS Contact: oasis-brains@nrg.wustl.edu.

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
