# Peer review of "Optimizing a 3D convolutional neural network to detect Alzheimer’s disease based on MRI"

_PeerJ Computer Science, doi:10.7717/peerj-cs.3129_

## Round 0.1 · original submission · Major Revisions

Dear authors,
You are advised to critically respond to all comments point by point when preparing an updated version of the manuscript and while preparing for the rebuttal letter. Please address all comments/suggestions provided by reviewers, considering that these should be added to the new version of the manuscript.

From my assessment, the authors should be more explicit in the novelty of their contribution.

Kind regards,
PCoelho

Reviewer 1 ·

Basic reporting

The author should refer recent articles and should be included in literature.
Equation (1) description in not specified.
Dataset description and the features in the dataset is not specified.
No Novelty in the work.

Experimental design

Need more to work on this paper
Technically need to improve the paper
only basic process has been done.

Validity of the findings

Need to give more explanation for the findings.

Additional comments

Need to improve the quality of the paper

Reviewer 2 ·

Basic reporting

The manuscript is written in a clear and professional way. The technical terms are appropriate, and the overall presentation is comprehensible for an audience familiar with machine learning and medical imaging. The introduction effectively outlines the significance of early Alzheimer’s Disease (AD) detection and contextualizes the problem. Key hypotheses (e.g., Cholinergic and Amyloid Hypotheses) are briefly explained. However, the connection between these hypotheses and the need for better classification models could be expanded.

The review of existing studies is comprehensive and includes methods ranging from traditional machine learning to deep learning approaches. However, the literature review could benefit from a more critical analysis of gaps in prior work. For example, it would be useful to emphasize why the proposed model improves upon the reviewed methods.

Figures illustrating the model architecture and ROC curves are relevant and well-labeled. The classification report (Table 1) is clear, but Table 2 (Comparison with State-of-the-Art Methods) has redundant text and requires better formatting.

The authors mention that the OASIS dataset is publicly available, but there is no explicit confirmation of raw data accessibility for replication. A direct link to the dataset and preprocessing steps would strengthen the transparency of the study.

Experimental design

The study addresses an important issue within the journal’s scope. The research question—enhancing AD classification using a novel 3D-CNN—is relevant and fills a knowledge gap in the application of deep learning to neuroimaging. The methods are well documented, including preprocessing, data augmentation, and model architecture. Key details include resizing MRI images to 128x128x32 and using 50 epochs for training. While sufficient for reproducibility, the rationale for specific parameter choices (e.g., input dimensions and augmentation strategies) should be elaborated.

Validity of the findings

The classification report shows a high test accuracy of 91%. Metrics such as precision, recall, and F1 score are adequately presented, and no significant overfitting is observed. However, further statistical analysis, such as confidence intervals for accuracy, would add rigor to the results.

The model’s performance is compared with state-of-the-art methods. While the proposed model achieves competitive accuracy with reduced training time, its slightly lower performance (compared to AlexNet) should be discussed in the context of trade-offs (e.g., speed vs. accuracy). Details on preprocessing and augmentation are clear, but the lack of hyperparameter settings limits full reproducibility. Providing these details would enhance the study’s scientific value.

The conclusions are consistent with the findings, highlighting the model’s efficiency and potential for broader application. The authors’ suggestions for future work, such as exploring additional datasets and feature extraction methods, are appropriate and constructive.

Additional comments

The author is advised to look into the following:

1. Consider expanding on the connection between the hypotheses and the need for improved classification models to provide stronger context.

2. Add a more critical analysis of gaps in the prior work to highlight the novelty and necessity of your proposed method.

3. Reformat Table 2 for better clarity and conciseness, as the current presentation of the table has redundant text.

4. With respect to experimental design provide the rationale for specific parameter choices, such as input dimensions and augmentation strategies, to enhance reproducibility.

5. Include statistical analysis, such as confidence intervals, to add rigor and support for the reported metrics in the results or findings section.

Reviewer 3 ·

Basic reporting

Manuscript Title “Optimizing a 3D convolutional neural network to detect alzheimer's disease based on MRI (#108772)”

1) The author could include a proposed diagram to enhance the clarity and understanding of the proposed method.
2) Improve the resolution and clarity of the images.

Experimental design

3) Check the captions of Figure 4 and Figure 5.
4) The author has presented a comparison of the proposed model with AlexNet, GoogLeNet, and ResNet in Table 2. The author is advised to check and verify the number of layers.

Validity of the findings

5) There are a few repeated statements in the manuscript.
6) Provide the rationale for selecting an input size of 128×128×32. Discuss whether other sizes were tested and how this choice impacted performance.

---

## Round 0.2 · Minor Revisions

Dear authors,

Thanks a lot for your efforts to improve the manuscript.

Nevertheless, some concerns are still remaining that need to be addressed.
Like before, you are advised to critically respond to the remaining comments point by point when preparing a new version of the manuscript and while preparing for the rebuttal letter.

Kind regards,
PCoelho

Reviewer 1 ·

Basic reporting

Good

Experimental design

Good

Validity of the findings

Good

Reviewer 2 ·

Basic reporting

The manuscript is professionally well-written. The introduction provides a sound background, including a relevant discussion of Alzheimer's disease, the motivation for early detection, and hypotheses underpinning disease progression (e.g., cholinergic and amyloid hypotheses). The authors seem to have done a thorough literature review, covering a broad range of recent studies and establishing the context for the proposed work. The authors have provided relevant figures and in good quality. However, some figures (such as architecture diagrams) would benefit from clearer captions and annotations for reader interpretation. Table formatting is mostly adequate. References are appropriate and current, with the majority of citations falling between 2016–2025. All raw data appear to be based on the OASIS-3 dataset, and relevant metrics have been reported (accuracy, precision, recall, etc.).

Experimental design

The research fits well within the scope of the journal and provides a valuable contribution by leveraging a 3D CNN architecture optimised for Alzheimer’s detection using MRI. The study employs an enhanced SqueezeNet-based model, SE-ResNeXt, and tries to attain computational efficiency. The experimental setup, including preprocessing (resizing, normalisation, augmentation), is adequately described. However, details about data splitting (train/validation/test), class balance, and statistical validation could be more rigorously explained—for instance, whether cross-validation was used and how random seed control was managed. The stated architecture, "SESENet18", provides a deeper technical breakdown and provides improved reproducibility.

Validity of the findings

The study reports a classification accuracy of 91%, which is competitive with prior state-of-the-art results. The authors also show a confusion matrix and derived metrics, namely precision, recall, specificity, and F1 score, along with confidence intervals and ROC curves. The results confirm the robustness of the performance claim. Importantly, the authors note that no significant overfitting was observed and that class imbalance was minimised. The authors have validated these claims with supporting visualisations. The comparison with prior models (AlexNet, ResNet, etc.) is a strength, demonstrating the proposed model's computational efficiency, although it falls just short of the highest accuracy reported (94.6%). The use of the well-known OASIS-3 dataset and the reported metrics show credibility to their findings. The author is just requested to include statistical significance testing (e.g., p-values or CI overlaps) to support performance comparisons with prior studies.

Additional comments

The strenghts of the article can be summarized as follows:
• The paper addresses an important healthcare challenge using deep learning.
• Strong literature review and contextual framing.
• Competitive model performance with high efficiency.
• Clear motivation for using 3D CNNs to avoid information loss from 2D projections.
However few suggestions for improvement can also be found below:
1. Include statistical comparisons for performance metrics.
2. Discuss ethical and clinical considerations for deploying such models in real-world scenarios.
Some minor issues are also noted below:
• Figure 2 and 3 captions should be more descriptive.
• Abbreviations like "SESENet18" should be expanded on first mention.

---

## Round 0.3 · accepted · Accept

Dear authors, we are pleased to verify that you meet the reviewer's valuable feedback to improve your research.

Thank you for considering PeerJ Computer Science and submitting your work.

Kind regards
PCoelho